# The tumor therapy landscape of synthetic lethality

Biyu Zhang[1,4], Chen Tang[1,4], Yanli Yao[2,4], Xiaohan Chen[1], Chi Zhou [1], Zhiting Wei[1], Feiyang Xing[1], Lan Chen[2], Xiang Cai[3], Zhiyuan Zhang[2], Shuyang Sun [2✉] & Qi Liu [1✉]

Synthetic lethality is emerging as an important cancer therapeutic paradigm, while the comprehensive selective treatment opportunities for various tumors have not yet been explored. We develop the Synthetic Lethality Knowledge Graph (SLKG), presenting the tumor therapy landscape of synthetic lethality (SL) and synthetic dosage lethality (SDL). SLKG integrates the large-scale entity of different tumors, drugs and drug targets by exploring a comprehensive set of SL and SDL pairs. The overall therapy landscape is prioritized to identify the best repurposable drug candidates and drug combinations with literature supports, in vitro pharmacologic evidence or clinical trial records. Finally, cladribine, an FDA-approved multiple sclerosis treatment drug, is selected and identified as a repurposable drug for treating melanoma with CDKN2A mutation by in vitro validation, serving as a demonstrating SLKG utility example for novel tumor therapy discovery. Collectively, SLKG forms the computational basis to uncover cancer-specific susceptibilities and therapy strategies based on the principle of synthetic lethality.

[1] Translational Medical Center for Stem Cell Therapy and Institute for Regenerative Medicine, Shanghai East Hospital, Bioinformatics Department, School of Life Sciences and Technology, Tongji University, Shanghai, China. [2] Department of Oral and Maxillofacial-Head Neck Oncology, Shanghai Ninth People's Hospital, College of Stomatology, Shanghai Jiao Tong University School of Medicine, Shanghai, China. [3] Alibaba Cloud, Hangzhou, China. [4] These authors contributed equally: Biyu Zhang, Chen Tang, Yanli Yao. ✉email: shuyangs@shsmu.edu.cn; qiliu@tongji.edu.cn

Synthetic lethality occurs when the inhibition of two genes is lethal while the inhibition of each single gene is not. It can be harnessed to selectively treat cancer by identifying inactive genes in a given cancer and targeting their synthetic lethal partners. For most cancer mutations caused by a loss-of-function, there are no targeted therapies available, and SL provides additional opportunities. The best-studied example of targeted therapies exploiting the synthetic lethality (SL) principle is the use of poly-ADP ribose polymerase (*PARP*) inhibitors in breast and ovarian cancer harboring mutations in BReast CAncer gene (*BRCA*). In this case, *BRCA* and *PARP* play vital roles in DNA homologous recombination repair in response to DNA damage, leading to the occurrence of tumorigenesis. Treatment of BRCA-deficient tumors with PARP inhibitors generally selectively kills the cancer cells in breast and ovarian cancer[1].

Basically, the general concept of "synthetic lethality" can be divided into two categories: (1) SL, which occurs between the loss-of-function mutations for tumor suppressor genes (TSGs) and their partner gene. This is a genetic interaction where combination of two mutations or more leads to cell death, whereas a single mutation in any of the genes does not. and (2) SDL, which occurs between the oncogene and its partner gene[2]. This is a genetic interaction where an overexpression of oncogene (Gene B) combined with the under-expression of its partner gene (Gene A) kills the tumor cell. (Fig. 1). Overall, SL and SDL provide important guidance for uncovering cancer-specific susceptibilities and identifying cancer-specific treatments.

Many largescale geneknockout studies using CRISPR screening and RNAi screening such as Project Score[3] and Project DRIVE[4] have presented a comprehensive catalog of essential genes related to certain phenotypes. Various methods have been proposed to identify SL and SDL interactions based on these data, however, the selective treatment opportunities for various tumors have not yet been explored. To this end, we developed the Synthetic Lethality Knowledge Graph (SLKG, https://www.slkg.net/), which presents the comprehensive tumor therapy landscape of SL and SDL from a drug repositioning perspective. SLKG integrates the large-scale entity of different tumors, drugs, and drug targets by exploring a comprehensive set of 19,987 SL and 3039 SDL pairs. By curation with a well-defined drug repositioning scoring schema, 155 and 88 best drug repositioning results were obtained according to the integrative analysis of SL or SDL with tumor and drug annotations, respectively, and most are supported by the literature, in vitro pharmacologic evidence or clinical trial records. In addition, 38 drug combinations were identified by mining the knowledge graph for tumor treatments with maximized therapeutic effects and reduced side effects. Finally, cladribine, which is a FDA-approved multiple sclerosis treatment drug, was selected and identified as a repurposable drug for treating melanoma with *CDKN2A* mutation by a comprehensive in vitro validation, serving as a demonstrating experimental protocol to utilize SLKG for novel tumor therapy discovery. Taking together, SLKG forms the computational basis for exploiting the tumor therapy landscape and repurposing known drugs based on the principle of SL to uncover cancer-specific susceptibilities.

## Results

**Uncover the tumor therapy landscape on the basis of SL/SDL.** The general framework of our study is outlined in Fig. 2. We obtained SL and SDL gene pairs with comprehensive annotations by searching the literature and databases spanning comprehensive data sources, such as SynLethDB[5], DRIVE DATA PORTAL[4], DepMap[6], and Daisy[7] (Fig. 2). The relationships between tumors and mutant genes were organized through the DisGeNET[8], COSMIC[9], ONGene[10], and TSGene[11] databases. By integrating

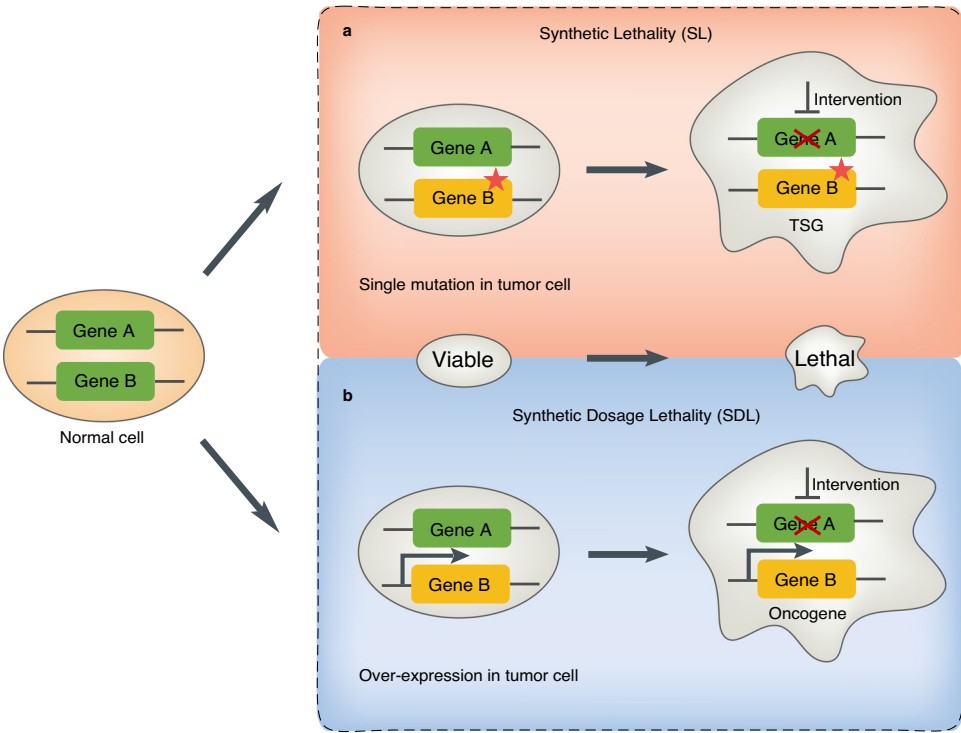

**Fig. 1 The principle of SL and SDL.** For gene A and B, if only one gene has a mutation or an overexpression in the tumor cell, the tumor cell is still alive. However, the pharmacological intervention of the partner gene will result in SL interaction in tumor cell which has a loss-of-function mutation of the tumor suppressor gene (TSG) (**a**). In addition, the pharmacological intervention of the partner gene will result in SDL interaction in tumor cell which has a gain-of-function mutation or an overexpression of the oncogene (**b**). The red star denotes a mutation. The thicker arrow denotes an overexpression. The cross line denotes a pharmacological intervention.

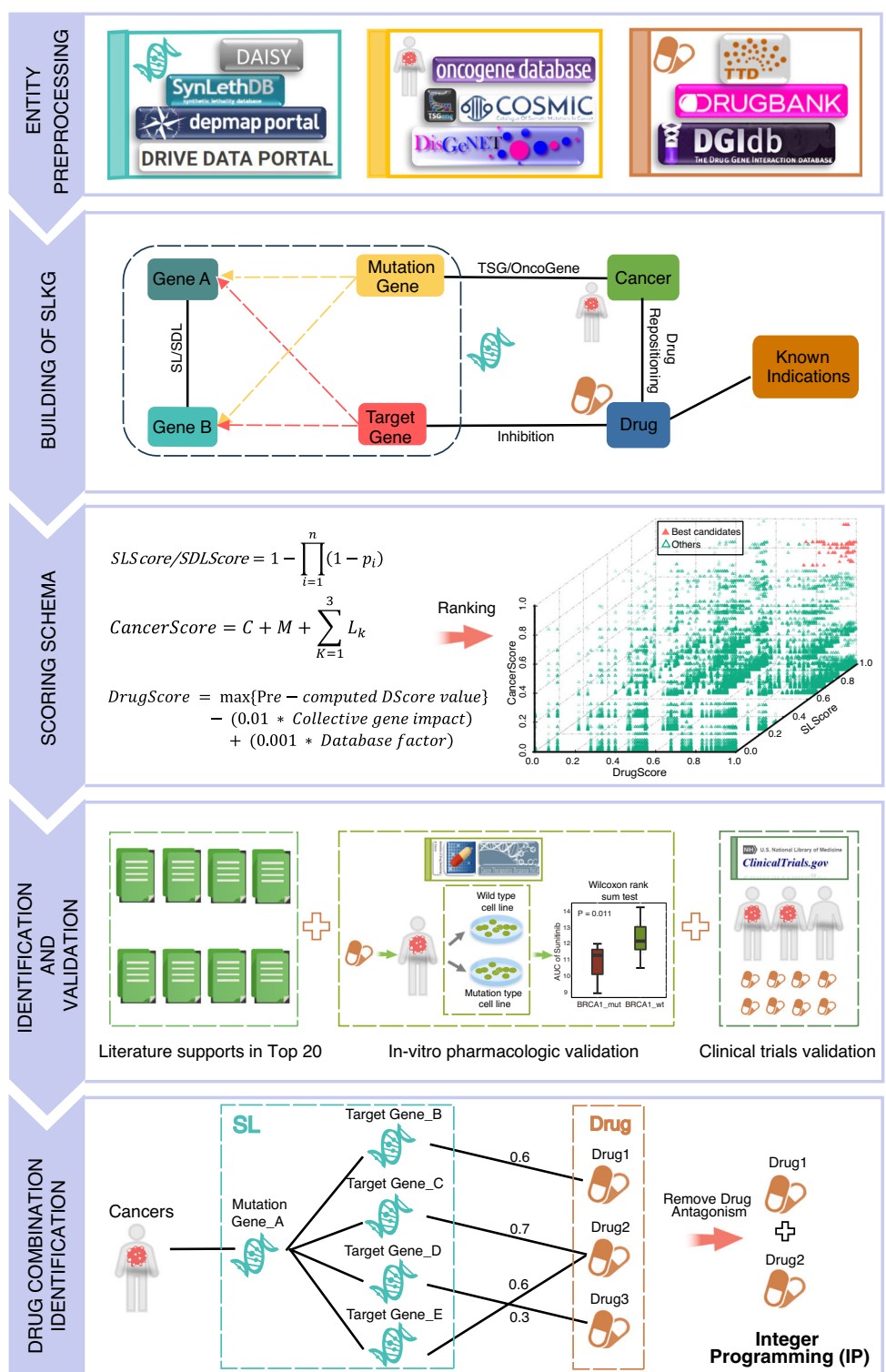

**Fig. 2 The general framework of the SLKG and the following analysis.** The study comprises five steps: (1) Entity preprocessing: the three types of entities (gene, cancer type, and drug) are collected and analyzed to exploit the tumor therapy landscape. (2) Building of SLKG: the SLKG comprises three types of entities and four types of relationships, which forms the computational basis from a drug repositioning perspective. (3) Scoring schema: a well-defined drug repositioning scoring schema by integrating three core scoring functions were developed to obtain the best repurposable drug candidates. (4) Identification and validation: the literature supports and in-vitro pharmacologic evidence were identified and validated in the top repurposable drug candidates. (5) Drug combination identification: a computational model was developed for identifying drug combinations in a weighted bipartite graph network by an IP algorithm.

the TTD[12], DrugBank[13], and DGIdb[14] databases, we further organized the relationships of drugs, target genes, and known indications into a knowledge graph. Based on these three types of entities, we mapped mutants and target genes to SL or SDL gene pairs, respectively. Because there is no obvious distinction between gene A and gene B in the SL or SDL pairs, we considered two reciprocal scenarios during the mapping, i.e., gene A maps the mutant gene and gene B maps the target gene, and vice versa

(see Methods). The relationship between the mutant gene and target gene was obtained on the basis of the relationship between gene A and gene B in the SL and SDL pairs. With SL or SDL as a bridge, the drug intervening with the target gene in SL or SDL was identified as a repurposable drug for treating a specific tumor with a mutant gene as the partner gene to the target gene in the SL or SDL (see Methods).

It should be noted that for either SL or SDL, the mode of action of the drugs on the target should be "inhibition". In addition, the mutant gene differs in SL and SDL, i.e., TSG for SL and oncogene for SDL[15,16]. Therefore, filtering by these criteria, 182,460 and 1830 drug repositioning results with drug name standardization and annotation of mutant genes were obtained according to the SL and SDL, respectively. All these curated data sources form the SLGK (https://www.slkg.net), which presents a comprehensive tumor therapy landscape based on SL and SDL.

**Identifying the top candidates based on a well-defined drug repositioning scoring schema.** By defining an effective drug repositioning scoring schema, the overall therapeutic landscape was prioritized to identify the top repurposable drug candidates. Specifically, three core scoring functions, i.e., the SLScore (SDLScore), DrugScore, and CancerScore, were defined by comprehensively considering the relationships between various entities in the SLGK. Min-max normalization was used to normalize the scores into [0, 1]. Based on the probability distribution curve and cumulative distribution curve of the three scores, a score threshold was set to obtain the best repurposable drug candidates by referring to the PanDrugs[17] database. Finally, 155 and 88 best repurposable drug candidates were obtained based on SL and SDL, respectively (Fig. 3a–d and Methods). A three-dimensional representation indicated that the best repurposable drug candidates were enriched in the upper right side with all three scores closer to 1 (Fig. 3e, f).

The robust rank aggregation[18] algorithm was further applied to sort the best repurposable drug candidates by integrating the ranks scored by the individual scoring function. Among the top 20 best repurposable drug candidates based on SL, 6 candidates were supported by the literatures[19,20] (Supplementary Table 1). For example, the third-ranked candidate indicates that rucaparib, a treatment for breast cancer, should have potential therapeutic effects on hereditary breast cancer and ovarian cancer syndrome, which was also reported in a previous study[19]. Similarly, among the top 20 best repurposable drug candidates based on SDL, 2 candidates were supported by the literatures[21,22] (Supplementary Table 1). For example, the sixth candidate indicates that vandetanib, a treatment for solid tumors, should have potential therapeutic effects for pancreatic ductal carcinoma, which is supported by in vitro findings[21].

**Validating the top candidates with in vitro pharmacologic evidence.** In addition to supporting information obtained from the literature, the drug sensitivity of cancer cell lines recorded in the CTRP[23] and GDSC[24] databases, whose measure indicators are mainly the half-maximal inhibitory concentration ($IC_{50}$) and the area under the curve (AUC), were exploited for top candidates validation. Smaller $IC_{50}$ and AUC values indicate that the cell line is more sensitive to the tested drugs[25]. Based on the cancer types provided by the Cancer Genome Atlas database (TCGA)[26], we obtained the cancer types of repurposed tumors (see Methods). The Wilcoxon rank sum test was applied to estimate whether the mutant type of a specific gene in the cancer cell lines was more sensitive to the drug than the wild-type, which indicates whether or not the drug has a potential therapeutic effect on the specific tumor. Our comprehensive explorations indicated that the top

significant difference in SL ($p = 0.011$) ranked 13th among the 155 best repurposable drug candidates. The median AUC of the BRCA1 mutant type cancer cell lines was smaller than that of the wild-type, indicating that an FLT3 inhibitor (sunitinib) has potential therapeutic effects for hereditary breast cancer and ovarian cancer with a BRCA1 mutation. Similarly, the top significant difference in SDL ($p = 0.034$) ranked 10th among the 88 best repurposable drug candidates. The median $IC_{50}$ of the NOTCH1 mutant type cancer cell lines was smaller than that of the wild-type, indicating that an EGFR inhibitor (gefitinib) has potential therapeutic effects for T-cell leukemia with a NOTCH1 mutation (Fig. 4a, b). Taking together, statistics on the in vitro pharmacologic data of cancer cell lines validated 12 SL-based repurposable drug candidates that account for 26% of the 47 candidates meeting the test criteria (Wilcoxon rank sum test under the AUC measurement, Fig. 4c and Supplementary Fig. 1a). Similarly, 6 SDL-based repurposable drug candidates were validated and account for 23% of the 26 candidates meeting the test criteria (Wilcoxon rank sum test under the $IC_{50}$ measurement, Fig. 4d).

Furthermore, top repurposable drug candidates were verified by clinical trial results in the ClinicalTrials.gov. Among the 88 best repurposable drug candidates based on SL, 10 of them were identified to be in clinical trials. Similarly, among the 48 best repurposable drug candidates based on SDL, 14 of them were identified to be in clinical trials. Collectively, it has shown that a substantial portion of the top repurposable drug candidates are registered in clinical trials, further proven the reliability of the prediction results (Supplementary Table 2).

**Drug combination identification by knowledge graph mining.** The biologic process of pathogenesis is usually diverse so that a single drug therapy that blocks one pathway often fails. It is reported that "synthetic lethality"-based therapy will also cause drug resistance and clinical recurrence to a certain extent due to alternative pathways. For example, homologous recombination can be restored through paralogues such as RAD51, PALB2, etc., in BRCA and PARP-deficient tumors[27]. Therefore, we further explored drug combination opportunities for tumor therapy with reduced resistance based on the SL and SDL mechanisms.

Specifically, we developed a computational model to identify the drug combinations based on the principle of SL and SDL by mining the SLGK with an IP algorithm (see Methods). We identified 38 potential drug combinations without antagonism between the 2 drugs (Supplementary Fig. 1b). For example, a combination of erlotinib and olaparib that covers four SL pairs, including *ABCG2 + BRCA1*, *BRCA1 + EGFR*, *BRCA1 + PARP1*, and *BRCA1 + PARP2*, which may have a potentially enhanced therapeutic effects with reduced resistance for hereditary breast cancer and ovarian cancer with the *BRCA1* mutation[28]. The known indications for erlotinib, whose target genes are *ABCG2* and *EGFR*, are colon cancer, glioma, head and neck cancer, etc. Similarly, the known indication for olaparib, whose target genes are *PARP1* and *PARP2*, is ovarian cancer (Fig. 5a).

An enrichment of the drug Anatomical Therapeutic Chemical (ATC) categorization is shown in Fig. 5b, indicating that these drugs tend to come from the ATC category L (antineoplastic and immunomodulating agents) and J (antiinfectives for systemic use). Then, a small number of drugs can be categorized into A (alimentary tract and metabolism), H (systemic hormonal preparations, excluding sex hormones and insulins), N (nervous system), S (sensory organs), and V (various). In addition, the repurposed cancer types for category L drugs are mostly enriched as the BRCA_OV, which are related to the mutant genes *BRCA1* and *BRCA2* (Fig. 5b). One possible reason for such enrichment is

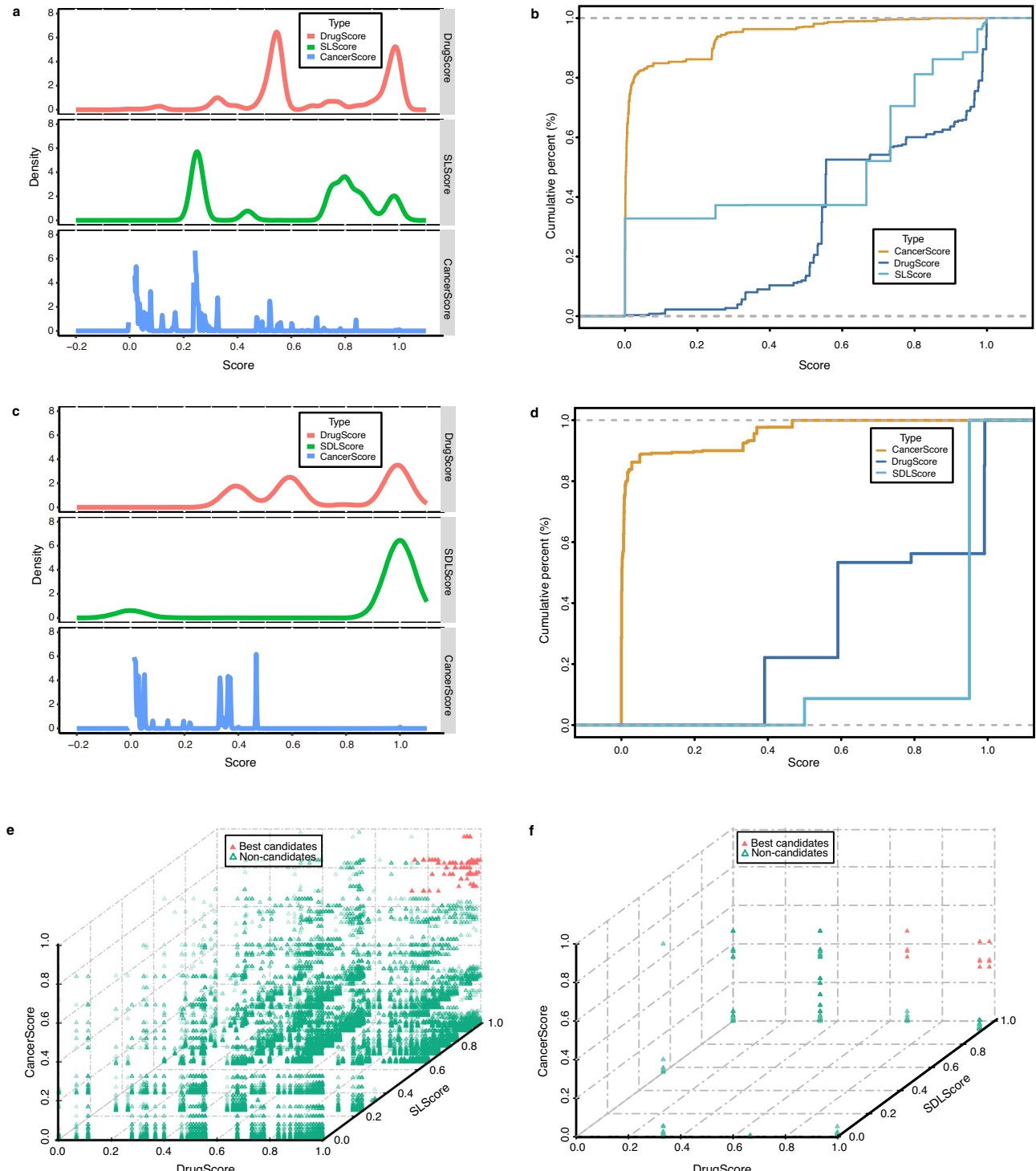

**Fig. 3 Distribution of three scores and top candidates. a** Probability distribution curve of the three scores in SL. Lines indicate DrugScore (red), SLScore (green), and CancerScore (blue). **b** Cumulative distribution curve of the three scores in SL. Lines indicate CancerScore (yellow), DrugScore (dark blue), and SLScore (sky blue). **c** Probability distribution curve of the three scores in SDL, same as SL. **d** Cumulative distribution curve of the three scores in SDL, same as SL. Three-dimensional graph of drug repositioning results for best repurposable drug candidates (red solid triangle) and non-candidates (green hollow triangle) based on SL (**e**) or SDL (**f**).

that the *BRCA* genes have been extensively studied in the literatures.

**A demonstrating in-vitro validation of cladribine to treat melanoma with CDKN2A mutation**. To further demonstrate the utility and validate the screening results of SLKG for novel tumor

therapy discovery, the top-ranked SL gene pair *CDKN2A + RRM2* (ranked 15th among the 155 curated SL) was selected as a demonstrating experimental validation for in vitro verification. The *CDKN2A* mutated melanoma cells are expected to be more sensitive to RRM2 inhibitor cladribine that was in clinical development for multiple sclerosis and leukemia, as reported by

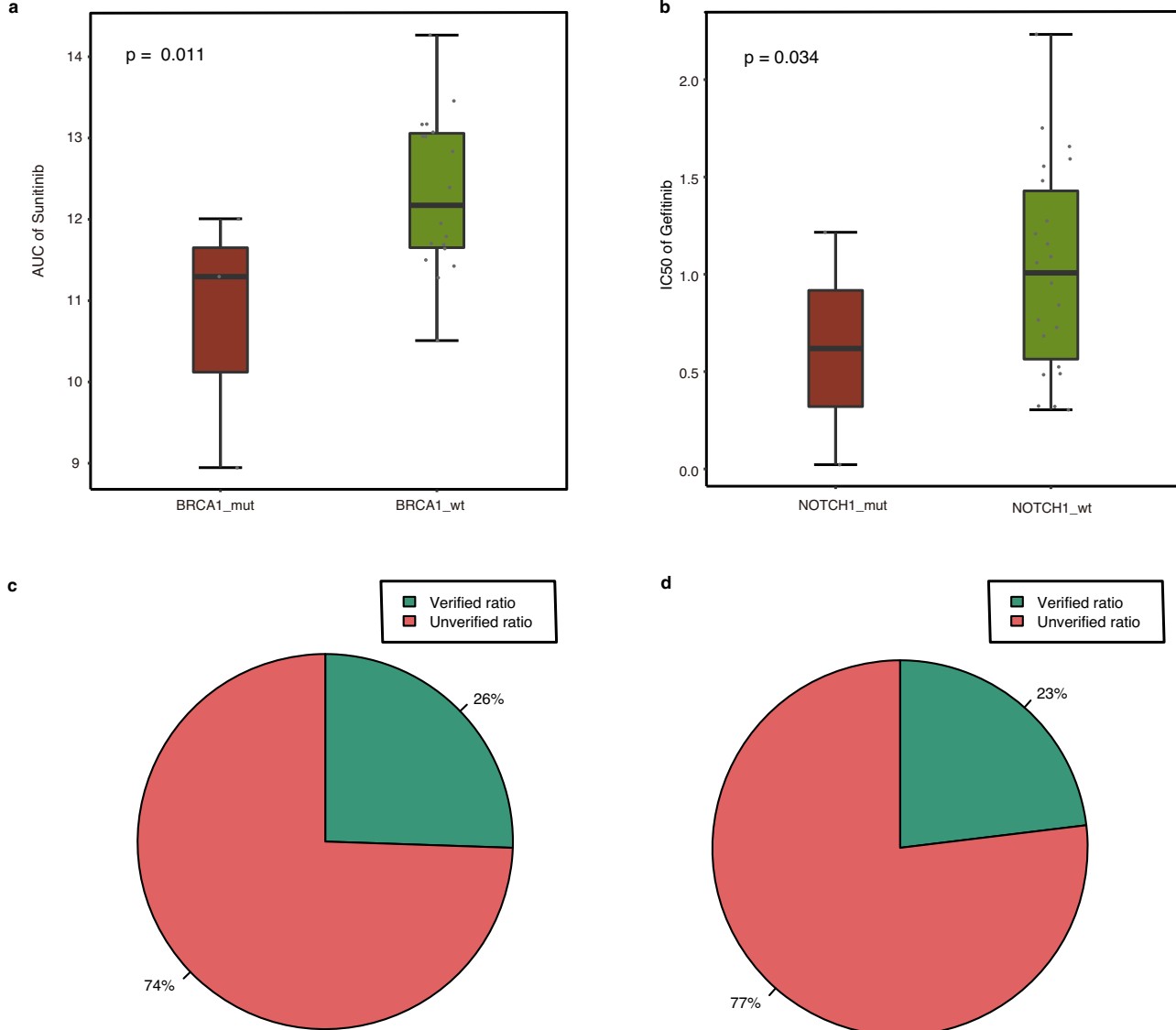

**Fig. 4 In vitro pharmacologic evidence.** Boxplot of the one-sided Wilcoxon rank sum test based on SL (AUC) (**a**) or SDL (IC$_{50}$) (**b**). Box plots indicate median (middle line), 25th, 75th percentile (box), and 5th and 95th percentile (whiskers) as well as outliers (single points), $n = 42$ independent experiments in SL (AUC) (**a**) and $n = 144$ independent experiments in SDL (IC$_{50}$); Dots represent cancer cell lines. Statistics of validations (green) and non-validations (red) in SL (**c**) or SDL (**d**).

SLKG. Previous study has indicated that *CDKN2A* is a key mutation for melanoma while the effective treatment of such mutation type of melanoma is still lack. In our study, six melanoma cell lines were treated with cladribine. Of the drug-cell line combinations, the melanoma cell lines exhibited diverse sensitivity to cladribine. Then IC50 values were used as parameters for drug potency. We compared drug sensitivity to *CDKN2A* mutations identified in melanoma cell lines. In the exon regions of *CDKN2A*, C32 and A375 cells showed nonsense mutations with *CDKN2A* loss of function (LOF), while M14, A2058, A875, and SK-MEL-1 were wild-type. We found consistent correlation between *CDKN2A* mutations and drug sensitivity. Compared with CDNK2A$^{WT}$ cell lines, CDKN2A$^{LOF}$ melanoma cell lines were more sensitive to cladribine (Fig. 6a).

Due to the different genetic backgrounds of melanoma cell lines, we adopted RNAi treatmet to knockdown *CDKN2A* expression in CDKN2AWT cell line A2058. Western blotting and RT-qPCR analysis were applied to assess the silencing

efficiency (Fig. 6b–d). As cell proliferation assays showed abrogating *CDKN2A* expression renders CDKN2AWT cells more sensitive to cladribine. Compared with their parental cell lines and the negative control cells, the A2058 lines with *CDKN2A* knockdown showed decreased cell viability after cladribine treatment, with cell viability that dropped from 50.85 to 29.20% (Fig. 6e).

## Discussion

Various methods have been proposed to identify "synthetic lethality" interactions. Selective treatment opportunities for all kinds of tumors have not yet been explored. To this end, we developed SLKG, which presents the comprehensive tumor therapy landscape of SL and SDL from a drug repositioning perspective. These results form a comprehensive reference for exploiting the tumor therapy landscape to uncover cancer-specific susceptibilities based on the principle of SL.

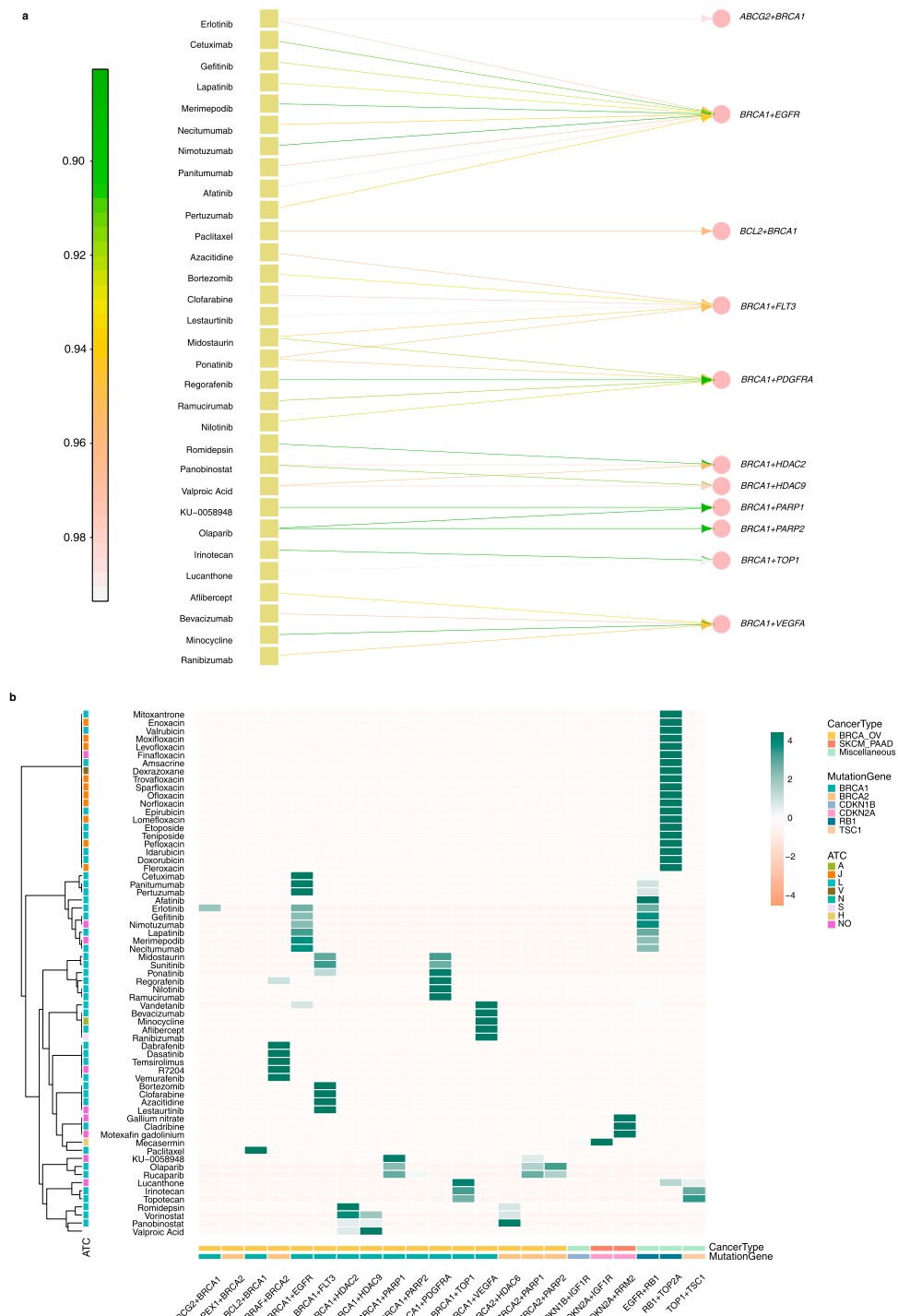

**Fig. 5 Example of drug combinations and the visualization of the drug combinations. a** Bipartite graph between drug (yellow square) and SL pairs (pink circle). Side annotation bar represents DrugScore. **b** Heat map visualization between drugs and cancer types of repurposed cancers. Side annotation bar represents the drug repositioning scores of each candidate.

Future updates and improvements are expected for this pioneer study including: (1) Standardizing the drug names for knowledge graph building. The drug names are inconsistent among databases. For example, the common name linsitinib is listed in DrugBank, but its trade name OSI-906 is listed in TTD. The current version of SLKG uses the PanDrugs drug naming system to unify the drug name across different datasets, but future standardization of the drug names is still needed. (2) Integrating the SL interaction and the synthetic dosage lethality interaction into one exploration. (3) Development of more sophisticated knowledge graph mining and link prediction algorithms to uncover the potential links between unconnected entities in SLKG, and (4) Collection of more pharmacologic evidence for repurposable candidate validation. Due to the limited data in GDSC and CTRP, the current proportion of validation for repurposable candidates is limited to in vitro data, collection of more pharmacologic evidence, especially in vivo experimental data, is required.

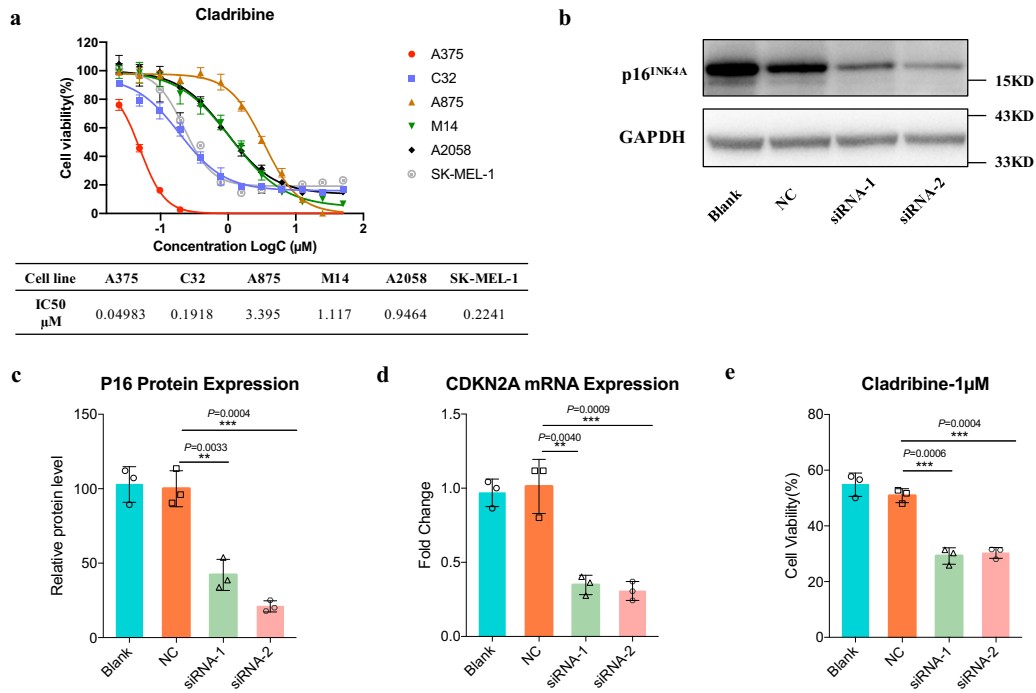

**Fig. 6 An in-vitro validation of cladribine treatment for melanoma cell lines with CDKN2A LOF mutations.** a *CDKN2A* LOF mutations in melanoma cell lines were correlated with sensitivity to *RRM2* inhibitors. Melanoma cell lines ($n = 6$) were incubated with 12 different concentrations of *RRM2* inhibitor cladribine and the IC50 values of cladribine were calculated (Data are means ± SD of the mean from three independent experiments). **b, c** *CDKN2A* silencing by siRNA in CDNK2A^WT cell A2058 and P16INK4A protein was measured by western blotting analysis ($n = 3$ biologically independent experiments, repeated at least 3 times independently with similar results; Mean ± SD shown; **$p < 0.01$, ***$p < 0.001$, two-tailed unpaired *t* test). **d** Relative mRNA expression was determined at 48 h post-transfection of *CDKN2A* siRNAs by RT-qPCR analysis ($n = 3$ biologically independent experiments; Mean ± SD shown; **$p < 0.01$, ***$p < 0.001$, two-tailed unpaired *t* test). **e** CDKN2A^LOF A2058 cells and corresponding control cell were treated with 1 μM cladribine measured in a 3-day in vitro growth assay ($n = 3$ biologically independent experiments; Mean ± SD shown; **$p < 0.01$, ***$p < 0.001$, two-tailed unpaired *t* test).

## Methods

**Building of SLKG**. The general architecture of SLKG and the following analysis is presented in Fig. 2. SLKG comprises three types of entities, gene, drug, and cancer type, and these three entities are linked with four types of relationships, (1) SL/SDL gene pairs, (2) the relationships between different cancer types and related mutant genes, (3) the relationships between drugs and their target genes, and (4) the relationships between drugs and their known indications (known cancer type).

*The entity and relationship construction of the SLKG.* The detailed entity collection and preprocessing of the SLKG is described as follows. First, the SL and SDL pairs were collected and curated by integrating the information from the SynLethDB, Daisy, DRIVE DATA PORTAL, and DepMap databases, as well as from the literatures. Second, DisGeNET v4.0 was applied to determine the cancer type according to the defined disease class, which is the leaf node in the disease ontology tree of the UMLS[29] database. After obtaining the primary relationship between the mutant genes and cancer types according to Concept Unique Identifiers, the mutant cancer genes provided by the COSMIC database, which is the authoritative resource for cancer research, were further used to screen the relationships between cancers and mutant genes. Finally, the relationships of drugs, target genes, and known indications were downloaded from three databases, i.e., the TTD database, the authoritative DrugBank database, and the DGIdb database of druggable targets. The relationships between target genes and drugs were analyzed according to the associations of the data in these databases, respectively. Four types of relationships were then curated on the basis of the relationships between the drugs and known indications in the TTD (Fig. 2: step 1).

After entity preprocessing of the SLKG, drug repositioning results were mapped and obtained. There are two reciprocal scenarios during the mapping, i.e., gene A maps the mutant gene and gene B maps the target gene, and vice versa. When the mutant gene is mapped to gene A, the target gene in the TTD, DrugBank, and DGIdb is mapped to gene B, respectively, and vice versa. It should be noted that in the DGIdb database, we must determine whether gene A or gene B is druggable firstly before mapping. To select the mode of action of the mutant gene, the TSGs and oncogenes were downloaded from the TSGene and ONGene databases, respectively. Based on the organized data of the three drug databases, the mode of action between the drugs and target genes was screened as "inhibition". Because the

drug names differ in these databases, they were first standardized by referring to the PanDrugs database during our data integration. The Identifier Exchange Service tool (https://pubchem.ncbi.nlm.nih.gov/idexchange/idexchange.cgi) in the PubChem database was applied to provide a full reference list of synonyms for specific compounds. We selected the first item of the returned synonyms as the standard drug name and manually modified the inconsistences. Moreover, to annotate the expression of mutant genes in cancer and adjacent cancer tissues, the URL link of each mutant gene in the GEPIA[30] database was provided. For example, the URL link (http://gepia.cancer-pku.cn/detail.php?gene=TP53) can directly jump to the page for TP53. On the annotation page, the Ensemble ID, gene name, and related summary information of TP53 are presented (Fig. 2: step 2).

*The drug repositioning scoring schema.* We developed an effective drug repositioning scoring schema by integrating three core scoring functions, including the SLScore (SDLScore), DrugScore, and CancerScore. Min-max normalization was used to normalize the three scores into [0, 1] (Fig. 2: step 3). First, a similar strategy was applied to the SynLethDB database to define the quantitative score $\rho_i$ to indicate the computational and experimental evidence supporting SL or SDL (Table 1). This experimental method is considered more reliable than the theoretical calculation with a higher quantitative score.

**Table 1 Quantitative score of SL/SDL.**

| Source of evidence | Type of evidence | $\rho_i$ |
|---|---|---|
| Biochemical experiment | CRISPR | 0.98 |
| | Synlethality | 0.95 |
| Related database | GenomeRNAi | 0.75 |
| | Decipher | 0.75 |
| Text mining | Text mining | 0.8 |
| Computational | Daisy | 0.5 |
| | Wang/Srihari/Ye/Srivas/Han et al. | 0.25 |

**Table 2 Pre-computed DScore value.**

| Drug status | Type of target gene | Pre-computed DScore value |
|---|---|---|
| Approved | Direct target | 1 |
| | Others | 0.8 |
| Clinical trials | Direct target | 0.6 |
| | Others | 0.4 |
| Experimental | Direct target | 0.2 |
| | Others | 0.1 |

Because the same SL or SDL pairs can be obtained in different ways, the SLScore (SDLScore) was calculated by integrating all the related evidences as follows:

$$s = 1 - \prod_{i=1}^{n} (1 - p_i) \tag{1}$$

where $n$ is the number of collection methods and $p_i$ is the quantitative score. For example, when the SL or SDL can be obtained through CRISPR and GenomeRNAi, the SLScore or SDLScore is 0.995 ($s = 1 - (1 - 0.98)(1 - 0.75)$).

It should be noted that the SLscore/SDLscore is designed to measure the reliability of SL/SDL from different types of evidences. The corresponding score is larger when more types of evidences are collected for the same SL/SDL. Compared to SL, SDL is relatively rarely studied at present, therefore, the evidences accumulated for SDL are rare compared to those of SL, which made SDLscore less discriminable compared to that of SLscore.

Second, with reference to the strategy applied in the PanDrugs database, the drug development status, drug specificity, and database number are integrated. Then, the DrugScore is defined as:

$$\text{DrugScore} = \max\{\text{Pre} - \text{computed DScore value}\}$$
$$- (0.01 \times \text{Collective gene impact})$$
$$+ (0.001 \times \text{Database factor}) \tag{2}$$

Where Pre-computed DScore value considers the drug development status and the type of target gene (Table 2), which are calculated as:

$$\text{Pre} - \text{computed DScore value} = [\text{Drug Status} + \text{Target Gene Type}] \tag{3}$$

Finally, the CancerScore is collected from the DisGeNET database. After obtaining the three core scoring functions, the min-max normalization conversion function is applied:

$$x' = \frac{x - \min(x)}{\max(x) - \min(x)} \tag{4}$$

To decide the threshold score of the best drug repositioning candidates, we further investigated the probability distribution curve and cumulative distribution curve of the three scoring functions. In the probability distribution curve of the drug repositioning results based on SL, the distribution of the DrugScore and SLScore was ideal in that it was enriched in the area with the higher score, while the CancerScore was enriched in the area with a lower score (Fig. 3a). In the cumulative distribution graph, the proportion with a CancerScore greater than 0.7 was ~1% and became stabilized (Fig. 3b). Consequently, the thresholds of the three scoring functions are listed as follows (Supplementary Table 3):

$$\text{SLScore } [0, 1] > 0.7$$
$$\text{DrugScore } [0, 1] > 0.8 \tag{5}$$
$$\text{CancerScore } [0, 1] > 0.7$$

Similarly, in the probability distribution curve of the drug repositioning results based on SDL, the distributions of the DrugScore and SDLScore were ideal in that they were enriched in the area with the higher score, while the CancerScore was still enriched in the area with the lower score (Fig. 3c). In the cumulative distribution graph, the proportion with a CancerScore greater than 0.3 was ~10% and became stabilized (Fig. 3d). Consequently, the thresholds of the three scoring functions are listed as follows (Supplementary Table 3)

$$\text{SDLScore } [0, 1] > 0.8$$
$$\text{DrugScore } [0, 1] > 0.5 \tag{6}$$
$$\text{CancerScore } [0, 1] > 0.3$$

*Development of the SLKG webserver.* The SLKG webserver is available at https://www.slkg.net/, developed by Bootstrap+Vue+Jquery+D3js, which makes the interface friendly access (Fig. 7a). The backbone of the knowledge graph architecture is constructed with Neo4j, which is a graph-based database platform powering knowledge graph building and following analysis. SLKG provides seven query modules based on SL and SDL, including searching for SL (SDL) pairs by gene symbol (Fig. 7b), searching for cancer type by mutation gene (Fig. 7c), searching for mutant gene by cancer type (Fig. 7d), searching for target gene by

drug name (Fig. 7e), searching for drug name by target gene (Fig. 7f), searching for repurposable drugs by cancer type (Fig. 7g), and searching for repurposed cancer types by drug (Fig. 7h). In addition, users can select various restrictions to filter the results based on their own demands. The search results can be downloaded for further study.

**In vitro pharmacologic evidence collection and validation of top repurposable drug candidates**. The in vitro pharmacogenomics data recorded in the GDSC and CTRP databases were downloaded for validation. The responses of 1067 cancer cell lines to 251 anticancer drugs are described in the GDSC with measurements of the $IC_{50}$ and AUC. The sensitivity data of 481 small molecule drugs in 664 cancer cell lines with measurement of the AUC were also obtained in previous study related to CTRP.

The names of the repurposed cancer types were standardized based on TCGA. For example, Ovarian Carcinoma corresponds to the OV (Ovarian serous cystadenocarcinoma) cancer type (Supplementary Table 4).

We further investigated collected cancer cell lines with the cancer types and mutant genes annotations provided in COSMIC. The cancer cell lines with a consistent mutant profile as indicated by the SL/SDL were denoted as mutant type, and otherwise as wild-type. The pharmacologic data of both mutant and wild-type cancer cell lines for the best repurposable drug candidates were selected and compared using the Wilcoxon rank sum test to determine whether the mutant type was more sensitive than that of the wild-type to the repositioned drug (Fig. 2: step 4).

Finally, by a comprehensive search of the clinical trial records, top repurposable drug candidates were further verified by clinical trial records. In our study, comprehensive clinical study data from ClinicalTrials.gov is investigated and utilized. The best repurposable drug candidates were selected and retrieved with existing clinical trial records in the ClinicalTrials.gov, further indicating whether the candidates have been gone into clinical trials (Fig. 2: step 4).

**Drug combination identification**. We provide a computational model for identifying drug combinations based on the principle of "synthetic lethality" in the SLKG via an IP algorithm, which was formulated as the mining of a weighted bipartite graph network with drugs as the nodes on the right side of the network and SLs as the nodes on the left side of the network (Fig. 2: step 5). We did not identify drug combinations for SDL due to the limited number of SDLs collected in the SLKG.

After constructing the weighted bipartite graph network, the IP algorithm was applied to identify the drug combinations that cover the most SL pairs of shared mutant genes by maximizing the summation of the corresponding DrugScore with the following optimization function:

$$\max Z = \sum_{i=1}^{n} \sum_{j=1}^{m} \left( d_i s_{ij} \right) \tag{7}$$

$$\text{Subject to} = \sum_{i=1}^{n} d_i = 2 \tag{8}$$

where $n$ represents the number of drugs; $m$ represents the number of SL pairs; $d_i$ is 1 or 0, corresponding to whether or not drug $i$ is selected, respectively; and $S_{ij}$ is the DrugScore for drug $i$ in SL $j$.

After solving the optimization function to obtain the optimal drug combinations, it was necessary to consider the antagonism between drug interactions. The DrugComb database was used to filter the antagonistic drug interactions with an "S" score lower than 5, where the "S" score is presented to evaluate the interaction of drug combinations at their $IC_{50}$[31].

It should be noted that the ATC Classification System is applied to categorize the drugs in our system. The ATC code comprises 7 digits and divides the drug into 5 levels, the first of which is an anatomic classification divided into 14 categories. We used the CID (Compound ID) to obtain the ATC classification of corresponding drugs annotated in the PubChem database.

**An in-vitro validation of cladribine to treat melanoma with CDKN2A mutation**
*Cell lines and materials.* The four human melanoma cell lines used, A375, C32, A2058, and SK-MEL-1, were obtained from American Type Culture Collection (Manassas, USA), which have been authenticated by the provider. A875 was derived from National Infrastructure of Cell Line Resource (Beijing, China) and M14 was derived from Mingzhou Biotechnology (Ningbo, China) which have been authenticated by us. All melanoma cell lines were cultured according to the manufacturer's protocol. RRM2 inhibitor Cladribine (S1199) was purchased from Selleck Chemicals (Texas, USA).

*Cell proliferation assays.* Cell Counting Kit-8 assay was performed in accordance with the manufacturer's recommendations by Beyotime Biotechnology (Shanghai, China). Briefly, 2000 cells per well in 96-well plates were untreated or treated with indicated doses of Cladribine and incubated for 48 or 72 h. Cell Counting Kit-8 reagent was added to each well and absorbance value was measured after 2 h incubation at 37 °C.

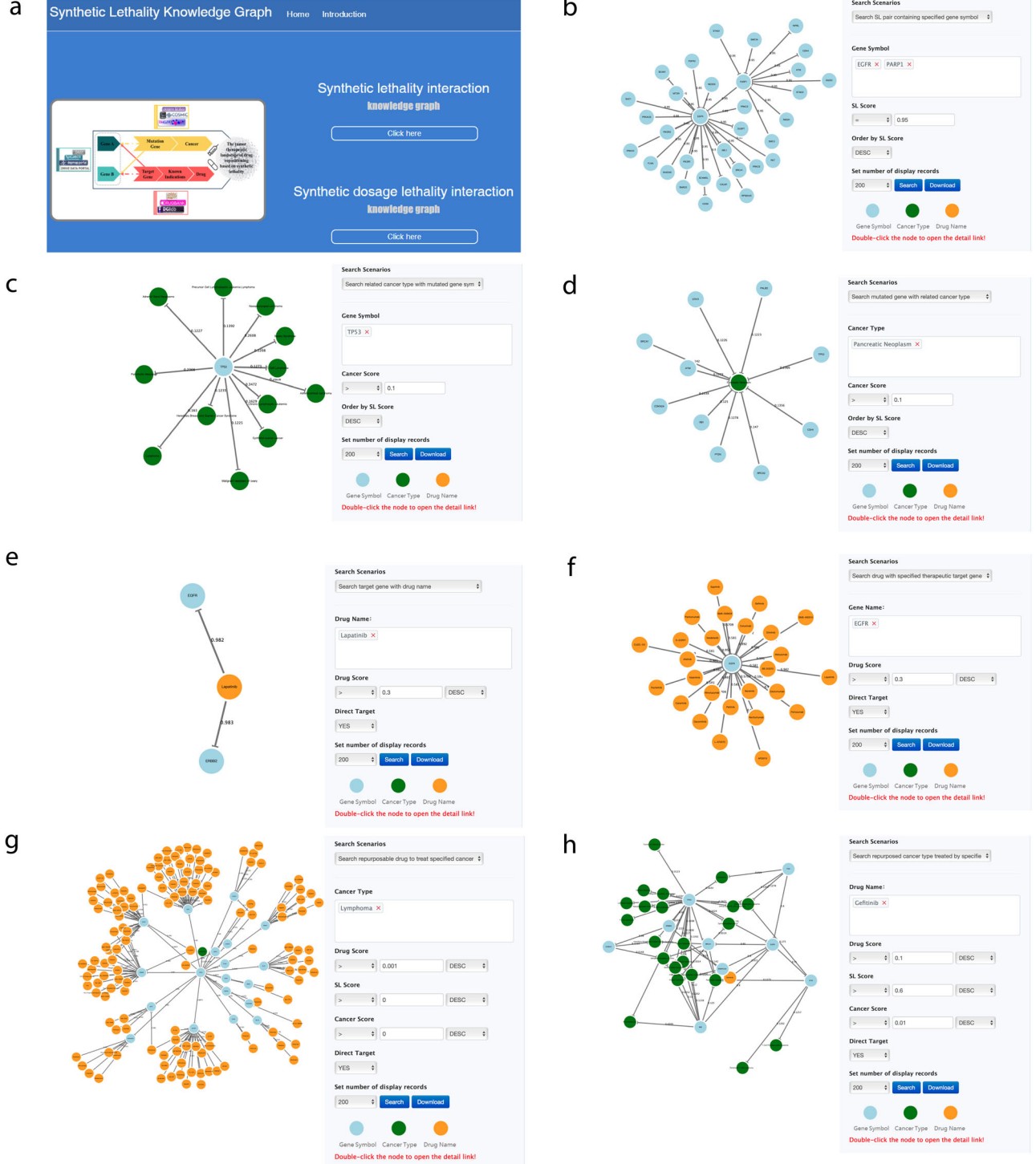

**Fig. 7 The query modes of SLKG webserver. a** Homepage of SLKG. **b–h** 7 query modules to explore the tumor therapy landscape based on the principle of "synthetic lethality".

*RNAi treatment.* For experiments involving siRNA-mediated depletion, two different *CDKN2A* siRNAs (siRNA1 and siRNA2) were designed and synthesized from GenePharma (Shanghai, China) with following targeted sequences: siRNA1, 5′- CACCAGAGGCAGUAACCAUU−3′, and siRNA2, 5′-CCCAACGCACCG AAUAGUUU−3′. The siRNA transfection was performed with Lipofectamine 3000 from Thermo Fisher (Massachusetts, USA) according to the manufacturer's instructions. The silencing efficiency was assessed by western blotting and reverse transcriptase-quantitative real time PCR (RT-qPCR) analysis. Antibodies used for western blotting including CDKN2A antibody (Cell Signaling Technology, #80772, 1:1000 dilution) and GAPDH antibody (Cell Signaling Technology, #5174, 1:1000 dilution). For quantification of western blotting, signal for each band was

quantified using ImageJ and normalized to loading control. Ratios of signal were calculated as a ratio of the relevant normalized signal quantifications.

**Reporting summary**. Further information on research design is available in the Nature Research Reporting Summary linked to this article.

## Data availability

SL and SDL gene pairs with comprehensive annotations are available from synlethDB and Daisy. The relationships between tumors and mutant genes are available from DisGeNET, COSMIC, ONGene, and TSGene. The relationships between drugs and

target genes are available from TTD, DrugBank, and DGIdb. The pharmacogenomics data for validation can be downloaded at GDSC and CTRP. Source data are provided with this paper. The Synthetic Lethality Knowledge Graph (SLKG) is available at https://www.slkg.net/.

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

## Acknowledgements

This work was supported by the National Key Research and Development Program of China (Grant No. 2017YFC0908500, No. 2016YFC1303205), National Natural Science Foundation of China (Grant No. 31970638, 61572361), Shanghai Natural Science Foundation Program (Grant No. 17ZR1449400), Shanghai Artificial Intelligence Technology Standard Project (Grant No. 19DZ2200900) and Fundamental Research Funds for the Central Universities.

## Author contributions

Q.L., S.Y.S., and Z.Y.Z. conceived the study. B.Y.Z. and C.T. performed the analysis and developed the Synthetic Lethality knowledge graph SLKG. C.T., X.H.C, Z.Y.Z, Z.T.W., X.C, and F.Y.X. processed the data. Y.Y.L. and L.C. performed the in-vitro experimental validation. Q.L., B.Y.Z., C.T., Y.Y.L., and S.Y.S. wrote the manuscript with assistance from other authors.

## Competing interests

The authors declare no competing interests.
