## [Peer Review File · Nature Communications]

Reviewers' Comments:

Reviewer #1:

Remarks to the Author:

This paper presented the first comprehensive tumor therapy resource of synthetic lethality and synthetic dosage lethality from a drug repositioning perspective. The scale of the resource is very big, including 19,987 synthetic lethality pairs and 3,039 synthetic dosage lethality pairs from many sources. There are other related resources, but having a one-stop shop with the sizable collection, drug repositioning ranking, and visualization tool is unique and valuable. Hence, it is a very useful resource for cancer drug development. The drug repositioning scoring schema is a little over-simplified and has room to improve, but for the database purpose, it may be sufficient. There is some experimental validation of the predictions. The paper is mostly well written. Overall this is an excellent study and should be published. But there is room for improvement.

1. Figure 1 can be more informative. There is little difference between the two graphs in the "Lethal" category.

2. Line 337: The way to integrate the data assumes that all the data are independent, which may not be the case. Some related limitation of SDLScore should be discussed. Ideally, a score with statistical meaning would be given, e.g. p-value, explainable probability, etc. But SDLScore doesn't have such meaning.

3. The website <http://www.slkg.net/> is preferably in the security mode (https). Otherwise, it is easy to get hacked. The system can be improved to be more user friendly. The "query" of the site shows the connecting genes to the queried gene, but the utility of the graph is limited. For example, it would be good to make the neighboring genes clickable to navigate to a new graph centered on the clicked gene. It is also desirable to show a table of the neighboring genes, indicating their functions, evidence source, reference, etc., where proper credits (papers and databases) should be acknowledged when applicable.

4. Some writing can be improved, for example:

Line 93: the literature supports and in-vitro pharmacologic evidence was identified ...  the literature supports and in-vitro pharmacologic evidence were identified

Line 446: An in-vitro validadation  An in-vitro validation

Reviewer #2:

Remarks to the Author:

The author present a knowledge based graph assembled by mining public accessible pharmacogenomic data repositories to the aim of providing a unified framework supporting the identification of drug repositioning opportunities for cancer.

Toward this aim the authors have designed a scoring system linking three types of entities (genes, diseases, drugs) through different type of relationships (synthetic lethal or synthetic dosage lethal gene pairs, drug-target links, cancer/frequently-mutatedGene) and assembled a computational infrastructure and a data portal for exploring the resulting knowledge based graph (through a dedicated user friendly interactive interface).

Briefly, knowledge base graphs are being increasingly used for drug discovery and to identify drug repositioning opportunities. They are are most frequently assembled via text mining of scientific literature. The idea of a knowledge based graph exclusively focusing on drug repurposing

exploiting the concept of synthetic lethality is quite original and timely given the increasing availability of public datasets from functional genetic screens (such as those performed within the Cancer Dependency Map partnerships).

In addition, the authors nicely show how their computational framework can be used to identify drug repositioning opportunities by selecting (and experimentally validating) cladribine as a potential therapy for melanomas with CDKN2A mutations.

If on one hand these aspects contribute in making this article of potential interest to the readers of Nature Communications, without implementing and offering to the user an automated strategy that mines the knowledge based graph and makes predictions by linking unconnected entities (potentially providing an integrated visualisation) the current framework and portal can be only considered a fancy way to explore/visualise relationships from public available datasets, which are weighted and summed together (within the same type of relations) and I am not sure that this is sufficient to warrant publication.

More worryingly, while testing the <http://www.slkg.net/> I noticed that the usual 'excel' date conversion issue affected some of the gene names. I think that this is unacceptable and it poses doubts on the quality of code/bioinformatic-pipelines underlying the resource presented in this manuscript.

In addition, the interface of the slkg portal only allows to explore trivial relationships without even attempting a single integrated visualisation where, for example, selecting a mutated gene, users see 'connected' cancer types, associated increased/decreased gene-dependencies and repurposable drugs (interesting by a clinical prospective) or seeding the search with a drug to see connected associated 'markers' and cancer types it could be repurposed, and exploited synthetic lethality. This is a pity as, the visualisation infrastructure seems to work fine.

Furthermore, the authors claim that they have 'repurposed' cladribine from multiple sclerosis to melanomas with CDKN2A losses. A drug can be said 'repurposed' when it reenters the market with a new indication, which is clearly not the case here. I would suggest to replace these claims with sentences like "we have identified cladribine as a repurposable drug for ..." or "we have identified cladribine as a drug repurposing opportunity for ..."

Finally, the author might have considered including synthetic lethal relations from project score (also part of the DepMap portfolio of datasets) among their 'processed entities' (<https://score.depmap.sanger.ac.uk/>)

Generally, the manuscript is not well written and clear enough. For example, the following sentence in the background section "For most cancer mutations caused by a loss-of-function, small molecules cannot be directly targeted" is unclear, maybe the authors wanted to say "For most cancer caused by loss-of-function mutations, there are no targeted therapies available" ?

Another example of a sentence that reads a bit odd "The best-studied example of targeted therapy according to the principle of SL is between poly-ADP ribose polymerase (PARP) and breast and ovarian cancer gene (BRCA)". This might be rewritten as "The best-studied example of targeted therapies exploiting the SL principle is the use of poly-ADP (...) in breast and ovarian cancers harbouring mutations in BRCA (...)"

A final minor point: The SDL acronym should be defined at its first occurrence in the main text, addition the SDL concept, i.e. genetic interaction whereby an underexpression of gene A combined with an overexpression of gene B kills the cell, should be defined and briefly discussed as the reader is left alone inferring this from figure 1. In addition the legend of figure 1 should be extended. Furthermore the use of '-' to indicate mutations might be misleading as it is traditionally associated with LoFs, and this is counterintuitive especially related to oncogenes. I would replace it

with `mut`

Reviewer #1:

This paper presented the first comprehensive tumor therapy resource of synthetic lethality and synthetic dosage lethality from a drug repositioning perspective. The scale of the resource is very big, including 19,987 synthetic lethality pairs and 3,039 synthetic dosage lethality pairs from many sources. There are other related resources, but having a one-stop shop with the sizable collection, drug repositioning ranking, and visualization tool is unique and valuable. Hence, it is a very useful resource for cancer drug development. The drug repositioning scoring schema is a little over-simplified and has room to improve, but for the database purpose, it may be sufficient. There is some experimental validation of the predictions. The paper is mostly well written. Overall this is an excellent study and should be published. But there is room for improvement.

Response:

Thanks a lot for this encouraging comment. We also made substantial revisions in this manuscript. Please see the following responses.

1. Figure 1 can be more informative. There is little difference between the two graphs in the “Lethal” category.

Response:

Thanks for this valuable comment. Fixed! Figure 1 was redrawn to be more comprehensive, especially to highlight the difference between SL and SDL.

Figure 1: The principle of SL and SDL. For gene A and B, if only one gene has a mutation or an over-expression

in the tumor cell, the tumor cell is still alive. However, the pharmacological intervention of the partner gene will result in SL interaction in tumor cell which has a loss-of-function mutation of the tumor suppressor gene (TSG) (part a). In addition, the pharmacological intervention of the partner gene will result in SDL interaction in tumor cell which has gain-of-function mutation or an over-expression of the oncogene (part b). The red star denotes a mutation. The thicker arrow denotes an over-expression. The cross line denotes a pharmacological intervention.

2. Line 337: The way to integrate the data assumes that all the data are independent, which may not be the case. Some related limitation of SDLScore should be discussed. Ideally, a score with statistical meaning would be given, e.g. p-value, explainable probability, etc. But SDLScore doesn't have such meaning.

Response:

Thanks for this comment. Sorry for not explaining this point clear. The SLscore/SDLscore is designed to measure the reliability of SL/SDL based on different types of evidences, either experimental or computational, and similar score schema was previously presented in SynLethDB. The corresponding score is larger when more types of evidences are collected for the same SL/SDL. Therefore, such score schema will not be affected by the data independence.

We agreed with the reviewer that the current version of SDLscore has certain limitations. Compared to SL, the main limitation of SDLscore lies in that SDL is relatively rarely studied at present, therefore, the evidences accumulated for SDL are rare compared to those of SL, which make SDLscore less discriminable compared to that of SLscore. Nevertheless, we believed that more experimental validated or computational evidence will be accumulated in the future for SDL, and we will keep to update the score schema in the future.

We have added a discussion on the score schema defined in our study in page 15 of "Method" section as:

"It should be noted that the SLscore/SDLscore is designed to measure the reliability of SL/SDL from different types of evidences. The corresponding score is larger when more types of evidences are collected for the same SL/SDL. Compared to SL, SDL is relatively rarely studied at present, therefore, the evidences accumulated for SDL are rare compared to those of SL, which make SDLscore less discriminable compared to that of SLscore. "

3. The website <http://www.slkg.net/> is preferably in the security mode (https). Otherwise, it is easy to get hacked. The system can be improved to be more user friendly. The "query" of the site shows the connecting genes to the queried gene, but the utility of the graph is limited. For example, it would be good to make the neighboring genes clickable to navigate to a new graph centered on the clicked gene. It is also desirable to show a table of the neighboring genes, indicating their functions, evidence source, reference, etc., where proper credits (papers and databases) should be acknowledged when applicable.

Response:

Thanks for these valuable suggestions. Fixed! The webserver is improved substantially in this version, including:

(1) Original HTTP mode is replaced by HTTPS as a security mode. Now the webserver can be accessed at <https://www.slkg.net/>. Also when the users visit the HTTP mode, it will be directly linked to HTTPS mode by port mapping.

(2) The search results are presented in a more user-friendly way with additional information. For all the 7 queries including “Searching for SL (SDL) pairs by gene symbol”, “Searching for cancer type by mutation gene”, “Searching for mutant gene by cancer type”, “Searching for target gene by drug name”, “Searching for drug name by target gene”, “Searching for repurposable drugs by cancer type”, and “searching for repurposed cancer types by drug”, a download button is presented to download the search results with comprehensive annotations. This result is stored in a EXCEL format for users to record and browse their searching results.

(3) The visualization of the last two queries, i.e., “Searching for repurposable drugs by cancer type” and “searching for repurposed cancer types by drug”, are modified to list all the related nodes (Maximum 200) and paths from the query cancer type to repurposable drugs, or vice versa. This is also related the comment of the 2nd reviewer, where an integrated relationship including all the related drugs, gene pairs and cancer types in the search results are included and visualized. We only modified the last 2 kinds of queries rather than all the 7 queries, since the last two queries are the two main focuses in our study, i.e., search repurposable drugs for certain cancer type or vice versa. We just leave other 5 kinds of queries to be unchanged in a concise way.

(4) A double click function is added for all 7 queries, helps users to directly access the detailed annotations of the drugs, genes and cancer types in the search results. The annotation databases for these three kinds of nodes are *PubChem*, *Genebank* and *DisGeNET* respectively. To this end, users can directly click each node they interested to see its detailed annotations.

4. Some writing can be improved, for example:

Line 93: the literature supports and in-vitro pharmacologic evidence was identified ... 

the literature supports and in-vitro pharmacologic evidence were identified

Line 446: An in-vitro validadation  An in-vitro validation

Response:

Thanks for this suggestion. Fixed!

Reviewer #2:

Briefly, knowledge base graphs are being increasingly used for drug discovery and to identify drug repositioning opportunities. They are most frequently assembled via text mining of scientific literature. The idea of a knowledge based graph exclusively focusing on drug repurposing exploiting the concept of synthetic lethality is quite original and timely given the increasing availability of public datasets from functional genetic screens (such as those performed within the Cancer Dependency Map partnerships).

In addition, the authors nicely show how their computational framework can be used to identify drug repositioning opportunities by selecting (and experimentally validating) cladribine as a potential therapy for melanomas with CDKN2A mutations.

Response:

Thanks a lot for this encourage comment.

1. While testing the <http://www.slkg.net/> I noticed that the usual 'excel' date conversion issue affected some of the gene names. I think that this is unacceptable and it poses doubts on the quality of code/bioinformatic-pipelines underlying the resource presented in this manuscript.

Response:

Thanks for this valuable suggestion. Fixed! This is due to the issue of "excel" date conversion. In this version, we fixed this issue and made a careful and comprehensive tests of the webserver to avoid such issues.

2. In addition, the interface of the slkg portal only allows to explore trivial relationships without even attempting a single integrated visualisation where, for example, selecting a mutated gene, users che see 'connected' cancer types, associated increased/decreased gene-dependencies and repurposable drugs (interesting by a clinicial prospective) or seeding the search with a drug to see connected associated 'markers' and cancer types it could be repurposed, and exploited synthetic lethality. This is a pity as, the visualisation infrastructure seems to work fine.

Response:

Thanks for these valuable comments. We agree with reviewer that an integrated and more user-friendly visualization should be presented. In this version, the webserver is improved substantially according to the reviewers' comments, including:

(1) Original HTTP mode is replaced by HTTPS as a security mode. Now the webserver can be accessed at <https://www.slkg.net/>. Also when the users visit the HTTP mode, it will be directly linked to HTTPS mode by port mapping. (This is based on the 1st reviewer's comment)

(2) The search results are presented in a more user-friendly way with additional information. For all the 7 queries including “Searching for SL (SDL) pairs by gene symbol”, “Searching for cancer type by mutation gene”, “Searching for mutant gene by cancer type”, “Searching for target gene by drug name”, “Searching for drug name by target gene”, “Searching for repurposable drugs by cancer type”, and “searching for repurposed cancer types by drug”, a download button is presented to download the search results with comprehensive annotations. This result is stored in a EXCEL format for users to record and browse their searching results.

(3) The visualization of the last two queries, i.e., “Searching for repurposable drugs by cancer type” and “searching for repurposed cancer types by drug”, are modified to list all the related nodes (Maximum 200) and paths from the query cancer type to repurposable drugs, or vice versa. where the integrated relationships including all the related drugs, gene pairs and cancer types in the search results are presented and visualized. **Users can seed on any node (Drug, Gene or Cancer type) to explore all the other connected nodes and repurposing paths from drugs to cancer types or vice versa.** It should be noted that we only modified the last 2 kinds of queries rather than all the 7 queries, since the last two queries are the two main focuses in our study, i.e., search repurposable drugs for certain cancer type or vice versa. We just leave other 5 kinds of queries to be unchanged in a concise way. In summary, these 7 kinds of queries presented a complete and integrated search and visualization of the search results from various perspectives.

(4) A double click function is added for all 7 queries, helps the users to directly access the detailed annotations of the drugs, genes and cancer types in the search results by linking the nodes to their corresponding annotations in the databases. The annotation databases for these three kinds of nodes are *PubChem*, *Genebank* and *DisGeNET* respectively. **To this end, users can directly click each node they interested to see its detailed annotations.**

3. Furthermore, the authors claim that they have ‘repurposed’ cladribine from multiple sclerosis to melanomas with CDKN2A losses. A drug can be said ‘repurposed’ when it reenters the market with a new indication, which is clearly not the case here. I would suggest to replace these claims with sentences like “we have identified cladribine as a repurposable drug for ...” or “we have identified cladribine as a drug repurposing opportunity for ...”

Response:

Thanks for this suggestion. Fixed! We also made other revisions throughout the manuscript.

4. The author might have considered including synthetic lethal relations from project score (also part of the DepMap portfolio of datasets) among their 'processed entities' (<https://score.depmap.sanger.ac.uk/>)

Response:

Thanks for this valuable comment. We checked Project Score carefully. Project Score is a wonderful data source that presented a quantitative measurement of the gene fitness related to certain phenotype by CRISPR screening, however, it does not provide validated synthetic lethal pairs directly. Nevertheless, the gene fitness data in this database can be utilized to discover SL/SDL interactions. We will keep the update by closely paying attention to this great data source. We also made a reference of this work in the last paragraph of “Background” section as:

“Many Large-scale gene-knockout studies using CRISPR screening and RNAi screening such as Project Score³ and Project DRIVE⁴ have presented a comprehensive catalogue of essential genes related to certain phenotypes.”

5. Generally, the manuscript is not well written and clear enough. For example, the following sentence in the background section “For most cancer mutations caused by a loss-of-function, small molecules cannot be directly targeted” is unclear, maybe the authors wanted to say “For most cancer caused by loss-of-function mutations, there are no targeted therapies available”?

Another example of a sentence that reads a bit odd “The best-studied example of targeted therapy according to the principle of SL is between poly-ADP ribose polymerase (PARP) and breast and ovarian cancer gene (BRCA)”. This might be rewritten as “The best-studied example of targeted therapies exploiting the SL principle is the use of poly-ADP (...) in breast and ovarian cancers harbouring

mutations in BRCA (...)”

Response:

Thanks for this suggestion. Fixed! We made a substantial revision of the manuscript in this version to make it clearer.

6. A final minor point: The SDL acronym should be defined at its first occurrence in the main text, addition the SDL concept, i.e. genetic interaction whereby an underexpression of gene A combined with an overexpression of gene B kills the cell, should be defined and briefly discussed as the reader is left alone inferring this from figure 1. In addition the legend of figure 1 should be extended.

Furthermore the use of ‘-’ to indicate mutations might be misleading as it is

traditionally associated with LoFs, and this is counterintuitive especially related to oncogenes. I would replace it with 'mut'

Response:

Thanks for those valuable comment. Fixed! We have defined the concept of SL and SDL at their first occurrences in the main text. In addition, Figure 1 was redrawn to avoid misleading based on the comments.

Figure 1: The principle of SL and SDL. For gene A and B, if only one gene has a mutation or an over-expression in the tumor cell, the tumor cell is still alive. However, the pharmacological intervention of the partner gene will result in SL interaction in tumor cell which has a loss-of-function mutation of the tumor suppressor gene (TSG) (part a). In addition, the pharmacological intervention of the partner gene will result in SDL interaction in tumor cell which has a gain-of-function mutation or an over-expression of the oncogene (part b). The red star denotes a mutation. The thicker arrow denotes an over-expression. The cross line denotes a pharmacological intervention.

The modified definition of SL and SDL is shown in Para 2 of Page 2:

“Basically, the general concept of “synthetic lethality” can be divided into 2 categories: (1) synthetic lethality (SL), which occurs between the loss-of-function mutations for tumor suppressor genes (TSGs) and its partner gene. This is a genetic interaction where combination of two mutations or more leads to cell death, whereas a single mutation in any of the genes does not. and (2) SDL, which occurs between the oncogene and its partner gene². This is a genetic interaction where an over-expression of oncogene (Gene B) combined with the under-expression of its partner gene (Gene A) kills the tumor cell. (Fig. 1).”

Reviewers' Comments:

Reviewer #1:

Remarks to the Author:

The authors made a great effort in revising the paper. Overall, it looks great.

The website is improved a lot. As possible future work (which doesn't need to be done in this paper), it would be nice to provide an option for users to integrate the synthetic lethality interaction knowledge graph and the synthetic dosage lethality interaction knowledge graph into one visualization with different node/edge colors for different types of interactions. In <https://www.slkg.net/introduction>, Figure 2's resolution is too low.

The writing can still be further improved, for example:

"In vivo", "in vitro", etc. should be in italic

Line 22:

a FDA-approved multiple  an FDA-approved multiple

Line 44:

suppressor genes (TSGs) and its partner  suppressor genes (TSGs) and their partner

Line 105:

3 core scoring functions was developed  3 core scoring functions were developed

Line 233:

CDKN2A mutated melanoma cells is  CDKN2A mutated melanoma cells are

Line 438:

indicating that whether the candidates have been
 indicating whether the candidates have been

Line 446:

The 233 CDKN2A mutated melanoma cells is expected to be
 The 233 CDKN2A mutated melanoma cells are expected to be

Reviewer #2:

Remarks to the Author:

The author have sufficiently addressed some of the points I raised in the previous round of revision. Nevertheless in my opinion, the lack of an automated strategy mining the knowledge based graph and making predictions linking unconnected entities makes the current framework and (as written before) the presented portal just a slightly more fancy way to explore/visualise binary relationships from public available datasets, which are weighted and summed together (within the same type of relations).

I do believe that this make this manuscript more suitable for a more specialised journal with a focus on scientific data or databases.

This aspect has not been addressed at all by the authors and my previous on this regard has not even been reported in their response letter, which I find quite unacceptable.

The portal is now accessible through a slightly more user-friendly GUI but, again, only trivial 1to1 relations can be explored and the authors did not even attempt implementing a single integrated visualisation of different network layers. As pointed also by reviewer n.1 the user

cannot 'expand' any node or gain additional infos. This makes this resources of limited impact and interests for the community. As written before, the visualisation infrastructure seems to be agile and working fine. It is a pity that the authors did not fully exploit it.

Reviewer #1:

The website is improved a lot. As possible future work (which doesn't need to be done in this paper), it would be nice to provide an option for users to integrate the synthetic lethality interaction knowledge graph and the synthetic dosage lethality interaction knowledge graph into one visualization with different node/edge colors for different types of interactions. In <https://www.slkg.net/introduction>, Figure 2's resolution is too low.

Response:

Thanks a lot for this encouraging comment. All the figure resolution including Figure 2 is modified. We also updated our future updating of SLKG as discussed in the Discussion Parts in Page 14:

“Future updates and improvements are expected for this pioneer study including: (1) Standardizing the drug names for knowledge graph building. The drug names are inconsistent among databases. For example, the common name linsitinib is listed in DrugBank, but its trade name OSI-906 is listed in TTD. The current version of SLKG uses the PanDrugs drug naming system to unify the drug name across different datasets, but future standardization of the drug names is still needed. (2) Integrating the synthetic lethality interaction and the synthetic dosage lethality interaction into one exploration. (3) Development of more sophisticated knowledge graph mining and link prediction algorithms to uncover the potential links between unconnected entities in SLKG, and (4) Collection of more pharmacologic evidence for repurposable candidate validation. Due to the limited data in GDSC and CTRP, the current proportion of validation for repurposable candidates is limited to *in vitro* data, collection of more pharmacologic evidence, especially *in vivo* experimental data, is required.”

The writing can still be further improved, for example:

“In vivo” , “in vitro” , etc. should be in italic

Line 22:

a FDA-approved multiple  an FDA-approved multiple

Line 44:

suppressor genes (TSGs) and its partner  suppressor genes (TSGs) and their partner

Line 105:

3 core scoring functions was developed  3 core scoring functions were developed

Line 233:

CDKN2A mutated melanoma cells is  CDKN2A mutated melanoma cells are

Line 438:

indicating that whether the candidates have been

 indicating whether the candidates have been

Line 446:

The 233 CDKN2A mutated melanoma cells is expected to be

 The 233 CDKN2A mutated melanoma cells are expected to be

Response:

Thanks for this valuable comment. Fixed! The whole manuscript is also edited.

Reviewer #2:

The author have sufficiently addressed some of the points I raised in the previous round of revision. Nevertheless in my opinion, the lack of an automated strategy mining the knowledge based graph and making predictions linking unconnected entities makes the current framework and (as written before) the presented portal just a slightly more fancy way to explore/visualise binary relationships from public available datasets, which are weighted and summed together (within the same type of relations). I do believe that this make this manuscript more suitable for a more specialised journal with a focus on scientific data or databases. This aspect has not been addressed at all by the authors and my previous on this regard has not even been reported in their response letter, which I find quite unacceptable.

Response:

Thanks for this valuable comment. Sorry for not quite catching your point in our previous reply. Now we fully understand your comment. Yes, we agreed with the reviewer that developing more sophisticated graph mining algorithm to uncover the potential links between unconnected entities is an interesting and useful direction to maximize the value of SLKG. The current study of SLKG are mainly focused on the mining of relationship between heterogeneous while connected entities. We have developed an efficient scoring schema and an Integer Programming algorithm to identify drug and drug combinations for cancer therapy, nevertheless, mining unconnected entities is a future direction worthy to be addressed. We included this point as one of our future updates of SLKG as listed in the Discussion Parts in Page 14:

“Future updates and improvements are expected for this pioneer study including: (1)

Standardizing the drug names for knowledge graph building. The drug names are inconsistent among databases. For example, the common name linsitinib is listed in DrugBank, but its trade name OSI-906 is listed in TTD. The current version of SLKG uses the PanDrugs drug naming system to unify the drug name across different datasets, but future standardization of the drug names is still needed. (2) Integrating the synthetic lethality interaction and the synthetic dosage lethality interaction into one exploration. (3) Development of more sophisticated knowledge graph mining and link prediction algorithms to uncover the potential links between unconnected entities in SLKG, and (4) Collection of more pharmacologic evidence for repurposable candidate validation. Due to the limited data in GDSC and CTRP, the current proportion of validation for repurposable candidates is limited to *in vitro* data, collection of more pharmacologic evidence, especially *in vivo* experimental data, is required.”

The portal is now accessible through a slightly more user-friendly GUI but, again, only trivial 1to1 relations can be explored and the authors did not even attempt implementing a single integrated visualisation of different network layers. As pointed also by reviewer n.1 the user cannot 'expand' any node or gain additional infos. This makes this resources of limited impact and interests for the community. As written before, the visualisation infrastructure seems to be agile and working fine. It is a pity that the authors did not fully exploit it.

Response:

Thanks for this suggestion. We actually added two integrated visualization and query entrances among the 7 queries, i.e., “Searching for repurposable drugs by cancer type”, and “searching for repurposed cancer types by drug”. Among these two searching scenarios the integrated relationships including all the related drugs, gene pairs and cancer types in the search results are presented and visualized in an integrated way. **Users can seed on any node (Drug, Gene or Cancer type) to explore all the other connected nodes and repurposing paths from drugs to cancer types or vice versa.** Therefore, we think we have tried to visualize the multiple relationships between heterogeneous nodes in an integrated way.

It should be noted that we only presented these 2 kinds of queries in an integrated way rather than all the 7 queries, since these two queries are the two main focuses in our study, i.e., search repurposable drugs for certain cancer type or vice versa. We just leave other 5 kinds of queries to be unchanged in a concise way. It will be very messy to visualize all the related nodes in one searching scenario or in one field. Therefore, we just designed 7 query scenarios for 7 specific query requirements. Nevertheless, these 7 kinds of queries presented a complete and integrated search and visualization of the search results from various perspectives.